# Correlation between Stress and Anxiety to Viral Epidemics (SAVE) and Burnout among Korean Dental Hygienists during the COVID-19 Pandemic: A Cross-Sectional Study

**DOI:** 10.3390/ijerph19063668

**Published:** 2022-03-19

**Authors:** Seul-Ah Lee, Jung-Eun Park, Jong-Hwa Jang

**Affiliations:** 1Department of Oral Health, Graduate School of Health and Welfare, Dankook University, Chungnam 31116, Korea; ssseulah@naver.com; 2Department of Dental Hygiene, College of Health Science, Dankook University, Chungnam 31116, Korea; jepark@dankook.ac.kr

**Keywords:** COVID-19, anxiety, burnout, job stress, dental hygienist

## Abstract

This study aimed to investigate the correlations among Stress and Anxiety to Viral Epidemics (SAVE), job stress (JS), and burnout among Korean dental hygienists during the COVID-19 pandemic and to identify the moderating effect of JS. As a cross-sectional study, a self-reporting questionnaire was used to survey 204 clinical dental hygienists to measure the levels of SAVE, JS, and burnout, along with their demographic characteristics as the control variables. Pearson correlation analysis and hierarchical multiple regression analysis were performed to analyse the correlations among burnout, SAVE, and JS, including the moderating effect of JS. With education level and subjective health controlled, JS (β = 1.05, *p* < 0.001), SAVE (β = 0.69, *p* = 0.020) and the interaction between SAVE and JS (β = −0.93, *p* = 0.050) were identified as significant influencing factors of burnout. The adjusted explanatory power of the model was found to be 52.4%. In summary, both SAVE and JS were significant influencing factors of burnout among dental hygienists, while a moderating effect of JS was also identified. Therefore, it is necessary to create a work environment that can relieve SAVE and JS to reduce burnout among dental hygienists.

## 1. Introduction

An emerging infectious disease refers to a newly identified infectious disease that is unrelated to any existing nationally notifiable disease. Novel infectious diseases require patient isolation, epidemiological investigation, and disease control measures due to concerns regarding their seriousness and rapid transmission. Major emerging infectious diseases that have occurred in Korea since 2000 include severe acute respiratory syndrome (SARS), avian influenza (AI), pandemic influenza A (H1N1), Middle East respiratory syndrome (MERS), and coronavirus disease 2019 (COVID-19) [1]. Of these, COVID-19, which is classified as a class 1 infectious disease (emerging infectious disease syndrome), has been affecting all parts of the world since December 2019. COVID-19 is a respiratory syndrome with RNA virus as the pathogen, which is transmitted through droplets, physical contact, and air, resulting in symptoms such as fever, cough, pneumonia, etc. [2].

From March 2020, when COVID-19 was declared as a global pandemic, to the present time, the human race has faced uncertainty in responding to the infectious disease [3]. Under such circumstances, healthcare workers have been reported to experience increased emotional stress due to outbreaks of the infection at their workplaces [4]. It has also been found that outbreaks of such emerging infectious diseases can cause burnout and mental health problems owing to infection and stress among healthcare workers [5].

A dental hospital is a conducive environment for viral transmission to occur easily due to aerosol-generating and invasive treatments [6,7]. Aerosols generated during dental procedures have the potential to contaminate adjacent and distant sites [8]. Dental healthcare workers are exposed to the risk of viral infection through face-to-face communication, exposure to saliva, blood, and other body fluids, and handling of sharp dental instruments. These healthcare workers are exposed to aerosolised contaminants and airborne pathogens in the treatment room. In addition, since aerosols can stay suspended in the air for hours before settling on surrounding surfaces or entering the airway, viral infection can easily spread to larger areas via the aerosols exhaled from infected dental personnel or patients [9]. Particularly, dental treatments performed on asymptomatic patients can lead to cross infection due to droplets. Consequently, dental workers can become carriers of cross infection for not only COVID-19, but also other infectious diseases such as hepatitis B and human immunodeficiency virus (HIV) [10,11].

Healthcare workers experience a deep sense of anxiety and burden about becoming infected, while also fearing the transmission of infection to their family and acquaintances. They also face excessive discomfort in wearing personal protective equipment (PPE) and are stressed by the tremendous workload with unclear disease control guidelines [12,13]. Further, healthcare workers are also subject to a high level of fatigue and stress at work, which can have negative psychological and physiological effects [14]. When such stress persists, it can cause increased burnout that leads to a decline in organisational efficiency due to workers’ health problems, absenteeism, poor work performance, and dissatisfaction, which could ultimately be a major predictor of turnover intention [15]. There have been many prior studies in Korea and the rest of the world reporting anxiety and stress among healthcare workers during epidemics [16,17,18,19,20,21]. However, few studies have included dental hygienists [22,23].

The COVID-19 pandemic brought forth the need to analyse the factors associated with burnout, as perceived by dental hygienists. Accordingly, this study hypothesised that Stress and Anxiety to Viral Epidemics (SAVE) and job stress (JS) perceived by dental hygienists during their work are positively correlated to burnout. Subsequently, this study aimed to analyse the level of burnout, SAVE, and JS perceived by dental hygienists alongside their correlations, while identifying the moderating effect of JS on burnout.

## 2. Materials and Methods

### 2.1. Study Design and Ethical Consideration

The present study was a cross-sectional survey that analysed factors associated with burnout as perceived by dental hygienists with regard to the COVID-19 pandemic and the moderating effect of JS on burnout. The study was conducted in accordance with the Declaration of Helsinki and was approved by the Institutional Review Board (IRB) of Dankook University (IRB No: DKU 2021-10-006). The hypothesis model of the study is as shown in Figure 1.

### 2.2. Participants and Data Collection

The study recruited dental hygienists working in a dental hospital or clinic in Korea. The sample size was calculated using the G *power 3.1.9.7 software (Heinrich-Heine-University, Düsseldorf, Germany) [24]. Based on a significance level of 0.05, effect size of 0.15, statistical power of 0.90, and 16 predictor variables, the sample size was calculated to be 175. Considering dropouts, the study population was set to 205 participants, after which the Naver online URL was used to conduct the questionnaire survey (http://naver.me/GDc6lkfT, accessed on 21 October 2021). After excluding one questionnaire for missing data and insincere responses, data from 204 participants were included in the final analysis.

Data were collected between 20 September and 17 October 2021. The participants were informed about the objective of the study through an information sheet and were requested to provide their consent to participate before completing the structured questionnaire.

### 2.3. Variables

#### 2.3.1. Outcome Variable

Burnout was measured using a burnout measurement scale containing 20 items developed in a previous study [25], which was modified and supplemented to suit dental hygienists, after consultations with an oral health expert.

The items were graded on a five-point Likert scale (ranging from 1: strongly disagree to 5: strongly agree). Positive items (*n* = 0) were reverse scored to calculate the overall mean score, with a higher score indicating a higher level of burnout. The Cronbach’s α value, which indicates the internal consistency of the tool, was 0.85 in a previous study [26] and 0.90 in the present study.

#### 2.3.2. SAVE 

SAVE was measured using the SAVE-9 tool developed in previous studies [27]. This tool, which was developed to measure anxiety and stress due to the viral epidemic among healthcare workers, consists of nine items. The items were graded on a five-point Likert scale (ranging from 1: strongly disagree to 5: strongly agree) to calculate the overall mean score, with a higher score indicating more severe SAVE. The Cronbach’s α value, which indicates the internal consistency of the tool, was 0.80 and 0.79 in the previous study [27] and this study, respectively.

#### 2.3.3. JS

JS was measured using a total of 19 items from questionnaire surveys used in previous studies [28,29,30,31]. The items were graded on a five-point Likert scale (ranging from 1: strongly disagree to 5: strongly agree) to calculate the overall mean score, with a higher score indicating higher JS. The Cronbach’s α value of the tool was 0.81 in a previous study that performed reliability analysis [30] and 0.84 in the present study.

#### 2.3.4. General Characteristics

The general characteristics used as covariates comprised of 13 items, including demographic and occupational characteristics based on previous studies related to burnout [15,25,26,27]. Demographic characteristics consisted of sex, age, area of residence, highest education level, marital status, religion, and subjective health status. Occupational characteristics included employment type, average income, career experience in years, workplace, field of work, and frequency of turnover. Area of residence was divided into Seoul, Gyeonggi-do, Incheon, Chungcheong-do, and others. Education level was classified as college, university, and ≥graduate school. Marital status was classified as unmarried, married, and others, whereas the employment type was permanent employee, contract worker, part-time, or others. Average income was categorised under <2,000,000, 2,000,000–2,500,000, >2,500,000–3,000,000, and >3,000,000 won. Career experience was classified as <1, 1–3, >3–10, and >10 years. Workplace was classified into clinic, hospital, and others, whereas field of work included comprehensive work, periodontal/surgical, conservative/prosthetics, orthodontics, desk and counselling, and management and others. Frequency of turnover was divided into none, 1–2, 3–4, and ≥5 times.

### 2.4. Statistical Analysis

The collected data were analysed using SPSS (IBM SPSS Statistics 23.0 for Windows, SPSS Inc., Chicago, IL, USA). Descriptive statistics were obtained for the major variables, while the Shapiro–Wilk test results assured normality. Burnout according to general characteristics was analysed by the independent t-test and one-way ANOVA, while Duncan’s post hoc test was performed for multiple comparison analysis. Correlations among SAVE, JS, and burnout were analysed using Pearson’s correlation analysis, while influencing factors of burnout and the moderating effect of JS were analysed using hierarchical multiple regression analysis. The significance level was set to 0.05.

## 3. Results

### 3.1. Burnout According to General Characteristics

Burnout according to the general characteristics of dental hygienists is as shown in Table 1. Burnout was found to be lower among participants with the highest education level of ≥graduate school (2.68 ± 0.68) than among those attending college (3.11 ± 0.64) and university (3.08 ± 0.70) (*p* = 0.011). Moreover, participants with poorer subjective health showed higher burnout (*p* < 0.001). Aside from the highest education level and subjective health status, there were no significant differences in burnout according to general characteristics (*p* > 0.05).

### 3.2. Descriptive Statistics for SAVE, Job Stress, and Burnout

Descriptive statistics for SAVE, JS, and burnout among dental hygienists are as shown in Table 2. The maximum possible score for each variable is 5 points and the results showed 3.56 points for SAVE (above average), 3.16 points for JS (average), and 3.04 points for burnout (average).

### 3.3. Correlations among SAVE, JS, and Burnout

Correlations among SAVE, JS, and burnout of the dental hygienists are as shown in Table 3. SAVE showed a weak correlation with JS (r = 0.265) and burnout (r = 0.288), whereas JS showed a strong positive correlation with burnout (r = 0.637). Therefore, the higher the SAVE, the higher the JS and burnout. In particular, it was found that the higher the JS, the higher the burnout.

### 3.4. Factors Affecting Burnout in Dental Hygienists: Moderating Effect of JS

Table 4 and Table 5 show the results of a hierarchical regression analysis on factors affecting burnout of dental hygienists. Analyses were performed by inputting variables into regression models: education level and subjective health, which showed significant differences in burnout according to general characteristics, were included as control variables in Model 1; SAVE, in Model 2; JS, in Model 3; and the interaction between SAVE and JS, in Model 4. F variation analysis results showed that JS moderated the relationship between burnout and SAVE. Moreover, the Variance Inflation Factor (VIF) of all four models was ≤2.52, indicating no multicollinearity.

In Model 1, education level and subjective health were influencing variables of burnout, while the fitness of the model was statistically significant (F = 29.401, *p* < 0.001).

In Model 2, the influence of SAVE was analysed, with education level and subjective health as control variables. The results showed that the influence on burnout increased by 3.5% as compared to Model 1 (*p* = 0.001). Considering influence by variables, SAVE had an impact on burnout (t = 3.23, *p* = 0.001), indicating that burnout increased as SAVE increased.

Model 3 analysed whether JS had a moderating effect on the relationship between burnout and SAVE. The analysis results showed that the fitness of the model was significant (F = 44.068, *p* < 0.001) and the explanatory power increased by 18.6% as compared to Model 2 (*p* < 0.001), indicating that JS had a moderating effect on burnout. Moreover, the results showed a positive correlation between burnout and JS (β = 0.48), implying that burnout increased as JS increased.

Model 4 verified whether JS interacted with burnout and SAVE to have a moderating effect. The analysis results showed that the fitness of the model was significant (F = 37.905, *p* < 0.001) and the explanatory power increased by 0.9% as compared to Model 3 (*p* = 0.050), indicating that JS had a moderating effect on the relationship between burnout and SAVE.

Such findings demonstrate that burnout increased when SAVE and JS increased. Moreover, considering the relationship between the independent variable SAVE and the dependent variable burnout, a research hypothesis was proposed, according to which, JS has a moderating effect on the relationship between SAVE and burnout.

## 4. Discussion

Dental hygienists comprise an occupational group with a high risk of infection due to the close contact they have with patients and due to the aerosol-generating procedures that they use in their work. Emerging epidemics cause JS among healthcare workers, including dental hygienists, due to the increased workload, job intensity, and anxiety about the risk of infection [23]. These factors can make them lose motivation to work in such environments and, consequently, experience burnout due to physical and mental exhaustion [14]. This study was conducted to identify the relationship between SAVE and burnout among dental hygienists and the moderating effect of JS on burnout during the COVID-19 pandemic. Based on the analysis results above, the following discussion focuses on SAVE and JS, which were found to be significantly associated with burnout among dental hygienists. The level of burnout was highest among those with an associate degree and lowest among those with a master’s degree or higher, which was consistent with the results in a previous study by Kim [32] and a work environment study [33]. According to a study on the job satisfaction of dental hygienists [34], a higher education level is associated with a higher professional self-concept, showing that the former could influence an increase in professionalism about their field of work. Moreover, considering the report that burnout decreased with an increase in positive psychological capital [35], it is believed that boosting confidence and self-esteem can help in reducing the level of burnout. The results showed higher burnout levels when the subjective health status of the participants was poor, which could be attributed to their greater anxiety and sense of burden regarding infection arising from a reduced belief and confidence about their own health.

The mean scores for SAVE and JS show some differences from those in previous studies. The SAVE score drawn in an earlier study on healthcare workers [16] was higher than that drawn in the current study (3.69 vs. 3.56). This may be due to the fact that more healthcare workers come into direct contact with confirmed COVID-19 patients than do dental hygienists. The JS score drawn in a pre-pandemic study [30] was lower than that drawn in the current study (3.00 vs. 3.16). This may be explained by the increased anxiety about infection, burden of wearing personal protective equipment, and disinfection work. The mean score for burnout (3.04) was in a similar range to that demonstrated in pre-pandemic studies [36,37]. Even before the pandemic, it was determined that the reasons for high burnout could be attributed to the dental hygienists’ physical burnout from increased stress and fatigue caused by physical limitations associated with excessive workload, skeletomuscular symptoms, fatigue, and headache [35,36,38].

In this study, there were positive correlations among SAVE, JS, and burnout, which meant that JS and burnout increased as SAVE increased. A study by Bae et al. [39] identified that burnout was correlated with anxiety and post-traumatic stress among nurses during the COVID-19 pandemic. Such results are consistent with the findings of this study. With the prolonged COVID-19 pandemic, healthcare workers experienced higher JS due to increased workload, as well as anxiety regarding transmission of infection, new work environment, and fatigue due to patient needs [40]. Such factors increased burnout, leading to negative outcomes.

The risk of stress among healthcare workers was aggravated by (a) the increasing number of critically ill patients around the time when COVID-19 was declared a pandemic, (b) the emergence of variants of COVID-19, (c) an unpredictable prognosis, and (d) a prolonged disaster situation. [41]. This state of affairs applies not only to healthcare workers at the forefront of fighting the COVID-19 crisis, but also to dental personnel working in environments vulnerable to the risk of infection due to aerosol generation, invasive procedures, and droplet exposure.

Such long-term excessive JS can lead to the development of psychological and physical health problems. From an organisational perspective, it can eventually lead to turnover consequent to poorer performance, dissatisfaction, and decreased organisational efficiency. Individual turnovers in turn lead to a deterioration in the quality of healthcare services, adversely affecting organisational management [15]. In this context, SAVE acts as a contributing factor in developing JS, and high levels of SAVE and JS can cause an increase in burnout, which may result in decreased personal growth and lower organisational efficiency.

Meanwhile, appropriate evidence-based discussion on our findings is vague due to a lack of prior studies on burnout among dental hygienists during the COVID-19 pandemic. In a study on nurses [37], those with higher JS and fatigue showed higher burnout. Moreover, another study on burnout among dental hygienists before the COVID-19 pandemic [36] reported that JS was associated with turnover intention. Repeated burnout causes JS among members of clinics [36], and loss of motivation for work could appear as job dissatisfaction, causing a negative influence [39]. Therefore, efforts to relieve JS and burnout among dental hygienists are also needed.

Carmassi et al. [41] noted that support from family and friends can help overcome post-traumatic stress in the COVID-19 situation. Thus, psychological support in the workplace and within the family can have a positive effect on dental personnel working in a high-risk environment susceptible to catch infection. Moreover, job training and regular counselling through stress intervention will be of help in reducing dental hygienists’ JS. In dental clinics, efforts need to be made to create a positive and improved work environment via clear communication with dental hygienists in the current pandemic situation through division of work and infection control measures [5].

It has also been reported that healthcare workers in hospitals and clinics, who are trained in COVID-19 infection prevention and safety management regulations, show high compliance rates in infection control practices [23]. Therefore, there is a need to find ways to introduce a regular education program related to infection control and a related institutional certification scheme. Additionally at the national level, a standardised infection control system will have to be implemented by adopting pertinent policies [23,42]. To ensure an effective response to future epidemics, it is essential to prepare policies and educational programs related to new infectious diseases. Such measures will also help dental hygienists and healthcare workers better adapt to similar situations if they occur in the future.

In a study on COVID-19 healthcare workers, negative emotions such as SAVE, fatigue, and helplessness were found to be dominant in the beginning. However, coping strategies were developed afterward due to adaptation, reasonable thinking, and support from others. In addition, positive emotions including responsibilities assumed by specialists, and their personal growth were also found [5]. As examined above, in the early days of the declared pandemic, there was a period of confusion due to unfamiliar tasks, ambiguity of roles, and frequently changing national quarantine policies.

Under these circumstances, dental hygienists should assume full responsibility as public health workers and expend efforts to prevent cross-infection by actively performing infection control activities. Thorough infection prevention activities should be undertaken through organisational and individual involvement towards the creation of a safe environment for patients and dental health professionals in a pandemic situation.

This study is noteworthy as it was the first to identify that SAVE and JS are significant influencing factors of burnout among dental hygienists in Korea working in medical institutions during the COVID-19 pandemic. However, the study also has some limitations. First, because the study population included only some dental hygienists working in the capital region and Chungcheong-do, the findings cannot be generalised for all the dental hygienists in Korea. Therefore, additional studies with a broader study population are needed. Second, the tools used to measure SAVE and burnout were developed for all healthcare workers who responded to emerging infectious diseases; thus, the tools may have been insufficient in accounting for the occupational identity of dental hygienists. Therefore, it is necessary to research and develop tools with proven validity and reliability for measuring SAVE among dental hygienists. Third, there were limitations in identifying certain details about the participants, including their vaccination status, history of confirmed infection, and experience of self-isolation. Therefore, future studies are needed with multi-dimensional variables and factors related to emerging infectious diseases. 

## 5. Conclusions

The findings of this study were found to support the hypothesis that SAVE and JS are factors influencing burnout among dental hygienists during the COVID-19 pandemic. In other words, dental hygienists’ burnout was found to increase as their SAVE and JS increased, while JS was found to have a moderating effect on the relationship between SAVE and burnout.

Therefore, there is a need for programs that reduce JS, relieve anxiety, and promote mental health among dental hygienists during epidemics, such as the COVID-19 pandemic. Additionally, there is a need to establish safe and efficient work environments and provide active support for relieving JS among this category of healthcare workers.

## Figures and Tables

**Figure 1 ijerph-19-03668-f001:**
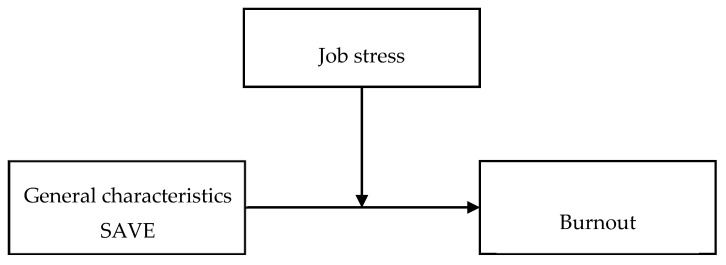
Hypothesis model of research. SAVE = Stress and Anxiety to Viral Epidemics.

**Table 1 ijerph-19-03668-t001:** Burnout according to the general characteristics of dental hygienists.

Variables	Division	*n* (%)	Burnout	t or F	*p*-Value (Duncan)
			Mean ± SD		
Sex	Male	21 (10.3)	2.98 ± 0.53	−0.411	0.682
	Female	183 (89.7)	3.05 ± 0.70		
Age (yrs)	<25	83 (40.7)	3.02 ± 0.68	0.089	0.915
	26–29	82 (40.2)	3.06 ± 0.73		
	≥30	39 (19.1)	3.03 ± 0.60		
Occupied area	Seoul	52 (25.5)	3.13 ± 0.67	0.478	0.752
	Gyeonggi-do	81 (39.7)	3.03 ± 0.71		
	Incheon	19 (9.3)	2.91 ± 0.69		
	Chungcheong-do	40 (19.6)	3.03 ± 0.67		
	Others	12 (5.9)	2.94 ± 0.62		
Education level	College	88 (43.1)	3.11 ± 0.64 ^a^	4.572	0.011
	University	88 (43.1)	3.08 ± 0.70 ^a^		(a > b)
	≥Graduate school	28 (13.7)	2.68 ± 0.68 ^b^		
Marital status	Unmarried	167 (81.9)	3.05 ± 0.71	0.229	0.796
	Married	35 (17.2)	2.98 ± 0.54		
	Others	2 (0.9)	3.24 ± 0.08		
Religion	Yes	53 (26.0)	2.91 ± 0.64	−1.557	0.121
	No	151 (74.0)	3.08 ± 0.69		
Subjective health	Very healthy	16 (7.8)	2.36 ± 0.62 ^a^	19.509	<0.001
	Healthy	46 (22.5)	2.62 ± 0.68 ^a^		(a < b < c)
	So-so	83 (40.7)	3.10 ± 0.53 ^b^		
	Unhealthy	54 (26.5)	3.44 ± 0.56 ^bc^		
	Very unhealthy	5 (2.5)	3.69 ± 0.59 ^c^		
Employment type	Permanent employee	168 (82.4)	3.02 ± 0.71	0.303	0.739
	Contract worker	27 (13.2)	3.13 ± 0.59		
	Part-time and others	9 (4.4)	3.05 ± 0.68		
Income (ten thousand won/month)	<200	22 (10.8)	3.05 ± 0.49	0.273	0.845
	200–250	90 (44.3)	3.07 ± 0.71		
	>250–300	69 (34.0)	2.98 ± 0.70		
	>300	22 (10.8)	3.05 ± 0.68		
Career (yrs)	<1	20 (9.9)	3.14 ± 0.73	0.291	0.832
	1–3	49 (24.1)	3.05 ± 0.68		
	>3–10	111 (54.7)	3.03 ± 0.70		
	>10	23 (11.3)	2.94 ± 0.60		
Workplace	Clinic	154 (75.5)	3.14 ± 0.73	1.994	0.116
	Hospitals	44 (21.6)	2.84 ± 0.59		
	Others	6 (2.9)	2.96 ± 0.38		
Type of work	Comprehensive work Periodontal, Surgical Conservative, Prosthetics	110 (53.9)	3.04 ± 0.68	0.745	0.591
		15 (7.4)	2.82 ± 0.60		
		44 (21.6)	3.10 ± 0.76		
	Orthodontics	9 (4.4)	3.20 ± 0.43		
	Desk and counselling	20 (9.8)	3.08 ± 0.62		
	Management and others	6 (2.9)	2.71 ± 0.78		
Frequency of turnover	None	52 (25.5)	2.90 ± 0.61	2.239	0.850
	1–2	89 (43.6)	3.00 ± 0.69		
	3–4	48 (23.5)	3.17 ± 0.74		
	≥5	15 (7.4)	3.31 ± 0.56		

SD = standard deviation; *p*-value was derived using the independent *t*-test or ANOVA test; a, b, c means followed by different letters are statistically significantly different at α = 0.05.

**Table 2 ijerph-19-03668-t002:** Descriptive statistics for SAVE, job stress, and burnout of dental hygienists.

Variables	Item	Range	Min	Max	Mean ± SD	Cronbach’s α
SAVE	9	1.00–5.00	1.44	5.00	3.56 ± 0.61	0.786
JS	19	1.00–5.00	1.47	4.42	3.16 ± 0.55	0.842
Burnout	20	1.00–5.00	1.29	4.88	3.04 ± 0.68	0.899

Min = minimum; Max = maximum; SD = standard deviation; SAVE = Stress and Anxiety to Viral Epidemics; JS = job stress.

**Table 3 ijerph-19-03668-t003:** Correlations among SAVE, JS, and burnout of dental hygienists.

Variables	SAVE	JS	Burnout
SAVE	1		
JS	0.265 **	1	
Burnout	0.288 **	0.637 **	1

** *p* < 0.01 by Pearson correlation analysis; SAVE = Stress and Anxiety to Viral Epidemics; JS = job stress.

**Table 4 ijerph-19-03668-t004:** Results of moderation effect using hierarchical regression analysis.

Model	R	R^2^	Adjusted R^2^	SE	Statistic Variation				
					R^2^	F	df1	df2	*p*-Value
1	0.555 ^a^	0.308	0.298	0.570	0.308	29.401	3	198	<0.001
2	0.586 ^b^	0.343	0.330	0.557	0.035	10.446	1	197	0.001
3	0.727 ^c^	0.529	0.517	0.473	0.186	77.493	1	196	<0.001
4	0.734 ^d^	0.538	0.524	0.470	0.009	3.890	1	195	0.050

Dependent variable: Burnout; D1 = college; D2 = university; SH = subjective health; SAVE = Stress and Anxiety to Viral Epidemics; JS = job stress. a. Predictors: (Constant), Education (D1), Education (D2), SH. b. Predictors: (Constant), Education (D1), Education (D2), SH, SAVE. c. Predictors: (constant), education (D1), education (D2), SH, SAVE, JS. d. Predictors: (constant), education (D1), education (D2), SH, SAVE, JS, SAVE * JS.

**Table 5 ijerph-19-03668-t005:** Hierarchical regression analysis of factors affecting burnout in dental hygienists.

Variables	Model 1			Model 2			Model 3			Model 4		
	β	t	*p*-Value	β	t	*p*-Value	β	t	*p*-Value	β	t	*p*-Value
Education (reference ≥ graduate school)												
D1	0.23	2.41	0.017	0.25	2.71	0.007	0.21	2.76	0.006	0.21	2.68	0.008
D2	0.30	3.23	0.001	0.29	3.20	0.002	0.24	3.09	0.002	0.23	3.02	0.003
SH	−0.53	−8.82	<0.001	−0.49	−8.10	<0.001	−0.32	−5.76	<0.001	−0.32	−5.93	<0.001
SAVE				0.19	3.23	0.001	0.10	1.93	0.055	0.69	2.28	0.020
JS							0.48	8.80	<0.001	1.05	3.57	<0.001
SAVE * JS										−0.93	−1.97	0.050
F (*p*-Value)		29.401 (<0.001)		25.714 (<0.001)			44.068 (<0.001)			37.905 (<0.001)		

Dependent variable = burnout; *p*-Value was derived using multiple regression analysis at α = 0.05; D1 = college; D2 = university; SH = subjective health; SAVE = Stress and Anxiety to Viral Epidemics; JS = job stress.

## Data Availability

The data presented in this study are available on reasonable request from the corresponding author.

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
