# Peer review of "Correlation between Stress and Anxiety to Viral Epidemics (SAVE) and Burnout among Korean Dental Hygienists during the COVID-19 Pandemic: A Cross-Sectional Study"

_ijerph, 2022, doi:10.3390/ijerph19063668_

Round 1
Reviewer 1 Report
The authors explored the correlations between Stress and Anxiety to Viral Epidemics (SAVE), job stress (JS), and burnout among Korean dental hygienists during the COVID-19 pandemic and investigated the moderating effect of JS on the relationship between SAVE and burnout. The topic of the study is relevant to International Journal of Environmental Research and Public Health. The study was well planned and executed. The manuscript was well written. However, the authors should address the following concerns before it can be accepted for publication.
1st Concern: On page 2, lines 80-81: The authors reported that they calculated the sample size based on “…a significance level of 0.05, effect of 0.15, statistical power of 0.90, and 17 predictor variables…” However, the authors reported that they investigated the effect of SAVE and JS on burnout. Although there were 13 variables covering a range of demographic and occupational characteristics, I could only find 15(=13+2) predictor variables in the study. Even the authors included SAVE*JS as a variable, there were only 16 variables. Please clarify what the “17 predictor variables” are.
2nd Concern: On page 5, Table 1: Some data in occupational characteristics did not add up to 204 [income: the total number of respondents was 203 (=22+90+69+202) and career: the total number of respondents was 203 (=20+49+111+23)]. Please check your data file.
3rd Concern: Some inconsistencies were observed between the results presented in the text and the results shown in tables.
- Line 99: The Cronbach’s alpha value of burnout was 0.89 in the present study. However, Table 2 shows that the Cronbach’s alpha of burnout was 0.899. Therefore, the one shown in line 99 should be 0.90 (rather than 0.89).
- Line 113: The Cronbach’s alpha value of JS was 0.81 in the present study. However, Table 2 shows that the Cronbach’s alpha of JS was 0.842. So, please check your results.
- Line 230: The authors stated that “…mean SAVE score among dental hygienists was 3.56 out of 5 points. However, Table 2 shows that the mean SAVE score was 3.58. So, please check your results.
- In Abstract, lines 19-20: The authors stated that “…SAVE (beta =0.12, p=0.020), and the interaction between SAVE and JS (beta = -0.10, p=0.050). However, when I looked at Model 4 in Table 5 (page 7), I found that the betas of SAVE and SAVE*JB were 0.69 and -0.93, respectively. Please clarify why these results were different from the ones shown in the Abstract.
4th Comment: Line 162: The authors stated that “SAVE showed a negative correlation with JS (r=0.265) and burnout (r=0.288)”. As these two correlation coefficients did not have a negative sign, I believe that the authors meant “SAVE showed a weak correlation with…” If that is the case, please correct the sentence in the revised documents.
5th Comment: Line 168: Please rewrite the sentence “The results from hierarchical regression analysis on the correlation between…” The sentence was confusing.
6th Comment: The “rho-value” in Tables 1, 4 and 5 should be “p-value”.
7th Comment: JB in Tables 4 and 5 should be “JS”.
8th Comment: Line 321: “…Disease Potal” should be “…Disease Portal”.
9th Comment: Line 348: The journal name “SAGE J.” was incorrect. It should be “INQUIRY”.
10th Comment: Line 351: Please add a full stop between “COVID-19” and “Korean J. Health…”
Author Response
Thank you for reviewing our research. Below are our responses to your comments and queries. We have incorporates all your recommendations into the revised manuscript. Our revised paper has been checked by a native English speaker (American Journal Experts).
The authors explored the correlations between Stress and Anxiety to Viral Epidemics (SAVE), job stress (JS), and burnout among Korean dental hygienists during the COVID-19 pandemic and investigated the moderating effect of JS on the relationship between SAVE and burnout. The topic of the study is relevant to International Journal of Environmental Research and Public Health. The study was well planned and executed. The manuscript was well written. However, the authors should address the following concerns before it can be accepted for publication.
1st Concern: On page 2, lines 80-81: The authors reported that they calculated the sample size based on “…a significance level of 0.05, effect of 0.15, statistical power of 0.90, and 17 predictor variables…” However, the authors reported that they investigated the effect of SAVE and JS on burnout. Although there were 13 variables covering a range of demographic and occupational characteristics, I could only find 15(=13+2) predictor variables in the study. Even the authors included SAVE*JS as a variable, there were only 16 variables. Please clarify what the “17 predictor variables” are.
Authors’ response 1:
- We thank you for pointing this out. We apologize for the error in the number of predictor variables. and The correction has been made to 16 variables and the appropriate number of samples. However, please understand that the actual number of participants remain the same.
2nd Concern: On page 5, Table 1: Some data in occupational characteristics did not add up to 204 [income: the total number of respondents was 203 (=22+90+69+202) and career: the total number of respondents was 203 (=20+49+111+23)]. Please check your data file.
Authors’ response 2:
- Thank you for pointing this out. The variables 'income and 'career' are the results of the analysis in which one non-respondent was excluded. We have added the effective percentage to the frequency.
3rd Concern: Some inconsistencies were observed between the results presented in the text and the results shown in tables.
Authors’ response 3:
- Thank you for mentioning this concern. We checked the inconsistencies between the text results and the table results and revised them.
Line 99: The Cronbach’s alpha value of burnout was 0.89 in the present study. However, Table 2 shows that the Cronbach’s alpha of burnout was 0.899. Therefore, the one shown in line 99 should be 0.90 (rather than 0.89).
Line 113: The Cronbach’s alpha value of JS was 0.81 in the present study. However, Table 2 shows that the Cronbach’s alpha of JS was 0.842. So, please check your results.
- Thank you for pointing this out. We have revised the Cronbach’s alpha of burnout to 0.90 in the text, and Cronbach’s alpha of JS to 0.85.
Line 230: The authors stated that “…mean SAVE score among dental hygienists was 3.56 out of 5 points. However, Table 2 shows that the mean SAVE score was 3.58. So, please check your results.
In Abstract, lines 19-20: The authors stated that “…SAVE (beta =0.12, p=0.020), and the interaction between SAVE and JS (beta = -0.10, p=0.050). However, when I looked at Model 4 in Table 5 (page 7), I found that the betas of SAVE and SAVE*JB were 0.69 and -0.93, respectively. Please clarify why these results were different from the ones shown in the Abstract.
- Our apologies for the mismatch between the two. We have revised the mean SAVE score of 3.56 in Table 2 to match the text. In addition, the minimum score has been revised to 1.44. We also revised the corresponding sentence in the abstract as follows.: ’ ....With education level and subjective health controlled, JS (β = 1.05, p < 0.001), SAVE (β = 0.69, p = 0.020), and the interaction between SAVE and JS (β = -0.93, p = 0.050) were identified as significant influencing factors of burnout....’
4th Comment: Line 162: The authors stated that “SAVE showed a negative correlation with JS (r=0.265) and burnout (r=0.288)”. As these two correlation coefficients did not have a negative sign, I believe that the authors meant “SAVE showed a weak correlation with…” If that is the case, please correct the sentence in the revised documents.
Authors’ response 4:
- We are truly sorry for the errors in the textual interpretation. We have revised the sentence as follows: ‘SAVE showed a weak correlation with JS (r = 0.265) and burnout (r = 0.288)’.
5th Comment: Line 168: Please rewrite the sentence “The results from hierarchical regression analysis on the correlation between…” The sentence was confusing.
Authors’ response 5:
- Thank you for suggesting a rephrasing of the sentence. We have revised it as follows: ‘Tables 4 and 5 present the results of a hierarchical multiple regression analysis on factors affecting burnout of dental hygienists.”
6th Comment: The “rho-value” in Tables 1, 4 and 5 should be “p-value”.
Authors’ response 6:
- Thank you for pointing this out. All content with errors in text have been revised to ‘p-value’.
7th Comment: JB in Tables 4 and 5 should be “JS”.
Authors’ response 7:
- Thank you for pointing this out. All content with errors in text have been revised to ‘JS’.
8th Comment: Line 321: “…Disease Potal” should be “…Disease Portal”.
Authors’ response 8:
- Thank you for pointing out the typo. We have corrected it in the sentence.
9th Comment: Line 348: The journal name “SAGE J.” was incorrect. It should be “INQUIRY”.
Authors’ response 9:
- Thank you for correcting us on this point. We have revised the journal name to INQUIRY.
10th Comment: Line 351: Please add a full stop between “COVID-19” and “Korean J. Health…”
Authors’ response 10:
- Thank you for pointing this out. We have added the period and revised the journal name as follows.: Korean J. Health Serv. Manag.”
We have incorporated all your recommendations and corrections in the revised manuscript. Please let us know if you want us to make any further modifications and we shall be glad to do so. Thank you very much.

Reviewer 2 Report
Dear Author,
Thank you very much for your paper. In this paper, the authors presented a study entitled “ Correlation between Stress and Anxiety to Viral Epidemics (SAVE) and Burnout among Korean Dental Hygienists during the COVID-19 Pandemic: Moderating Effect of Job Stress” with aim to analyse the level of burnout, SAVE, and JS perceived by dental hygienists alongside their correlations, while identifying the moderating effect of JS on burnout.
In general, the manuscript is very interesting and well-written. The topic is in line with the journal aim. The review appears correctly performed and written without logical or factual errors. Of course, the fear and anxiety of being infected were the main characters, especially in the early stages of the pandemic.
However, some corrections are required to improve the overall quality. An English-language review is required.
My recommendations are the following:
Please insert on the title and abstract the type of the study in order to be immediately understandable for the reader.
The introduction section need to be increased : “A dental hospital represents an environment where viral transmission can occur easily due to aerosol-generating and invasive treatments [6]. Particularly, dental treatments performed on asymptomatic patients can lead to cross infection due to droplets. Consequently, dental workers can become carriers of cross infection for not only COVID-19, but also other infectious diseases such as hepatitis B and human immunodeficiency virus (HIV) [7].”
Material and Methods: This section is clear and well-performed .
Conclusions are clear and reflect results
According to this Reviewer’s consideration, novelty and quality of the paper, publication of the present manuscript is recommended after minor revision.
Author Response
Thank you for reviewing our paper. Below are our responses to your comments and queries. We have incorporated all your recommendations into the revised manuscript. Our revised paper has also been checked by a native English speaker (American Journal Experts).
Dear Author,
Thank you very much for your paper. In this paper, the authors presented a study entitled “Correlation between Stress and Anxiety to Viral Epidemics (SAVE) and Burnout among Korean Dental Hygienists during the COVID-19 Pandemic: Moderating Effect of Job Stress” with aim to analyse the level of burnout, SAVE, and JS perceived by dental hygienists alongside their correlations, while identifying the moderating effect of JS on burnout. In general, the manuscript is very interesting and well-written. The topic is in line with the journal aim. The review appears correctly performed and written without logical or factual errors. Of course, the fear and anxiety of being infected were the main characters, especially in the early stages of the pandemic.
However, some corrections are required to improve the overall quality. An English-language review is required.My recommendations are the following: Please insert on the title and abstract the type of the study in order to be immediately understandable for the reader.
Authors’ response 1:
- Thank you for your valuable suggestion. The title has been revised to ‘Correlation between Stress and Anxiety to Viral Epidemics (SAVE) and Burnout among Korean Dental Hygienists during the COVID-19 Pandemic: A Cross-Sectional Study’. We have also added the type of study (Cross-Sectional Study) to the abstract.
The introduction section need to be increased : “A dental hospital represents an environment where viral transmission can occur easily due to aerosol-generating and invasive treatments [6]. Particularly, dental treatments performed on asymptomatic patients can lead to cross infection due to droplets. Consequently, dental workers can become carriers of cross infection for not only COVID-19, but also other infectious diseases such as hepatitis B and human immunodeficiency virus (HIV) [7].”
Authors’ response 2:
- Thank you for your suggestion. In response to your comment, We have added the following sentence: ‘Aerosols generated during dental procedures have the potential to contaminate adjacent and distant sites [8]. Dental healthcare workers are exposed to the risk of viral infection through face-to-face communication, exposure to saliva, blood, and other body fluids, and handling of sharp dental instruments. They are exposed to aerosolised contaminants and airborne pathogens in the treatment room. In addition, since aerosols can stay suspended in the air for hours before settling on surrounding surfaces or entering the airway, viral infection can easily spread to larger areas via the aerosols exhaled from infected dental personnel and patients [9]. ”
“........”.
Material and Methods: This section is clear and well-performed.
Conclusions are clear and reflect results
According to this Reviewer’s consideration, novelty and quality of the paper, publication of the present manuscript is recommended after minor revision.
We shall be glad to incorporate any further revisions required. Thank you very much.

Reviewer 3 Report
First of all, thank you for being able to read, evaluate and contribute something in your work Correlation between Stress and Anxiety before Viral Epidemics (SAVE) and Burnout among Korean dental hygienists during
COVID-19 pandemic: moderating effect of work stress
This study aimed to investigate the correlations between Stress and Anxiety to Viral Epidemics (SAVE), work stress (JS), and exhaustion among Korean dental hygienists during the COVID-19 pandemic to identify the moderating effect of JS. A survey questionnaire was used 204 clinical dental hygienists to measure SAVE, JS, and burnout levels, along with
its demographic characteristics such as control variables. Pearson correlation analysis and hieranalysis of multiple regression to analyze correlations between exhaustion, SAVE, and JS, including the moderating effect of JS. They were identified as significant factors influencing exhaustion. The explanatory power of the model was 52.4%. In summary, both SAVE and JS were significant burnout influence factors among dental hygienists, while a JS moderating effect was also identified. Therefore, it is necessary to create a working environment that can alleviate SAVE and JS to reduce exhaustion among dental hygienists.
It'is a good job, on a very interesting topic, but before being published you must incorporate the following suggestions, I ask the authors to follow them one by one,
In the introduction some important references are missing, revise it,
Although it is deduced from the text, it must explain the research gap that the article tries to cover,
The explanation of the procedure carried out for the investigation needs to be improved, it is unclear,
The section of statistical analysis, has to be worked more, much more
The discussion section must be improved and more worked, much more worked, there are hardly any lines, which are insufficient for a work to be published in our journal. In addition, you must connect with the introduction. This can be extended to the bibliography of the previous studies of the introduction that must be connected with the discussion, that will give more power to the manuscript,
the same with the conclusions, are excessively brief
With these changes made, the article will improve and will have the quality to be published in this prestigious journal
kind regards
Author Response
Thank you for reviewing our research. Below are our responses to your comments and queries. We have incorporated all your recommendations into the revised manuscript. Our revised paper has been checked by a native English speaker (American Journal Experts).
It'is a good job, on a very interesting topic, but before being published you must incorporate the following suggestions, I ask the authors to follow them one by one, In the introduction some important references are missing, revise it,Although it is deduced from the text, it must explain the research gap that the article tries to cover,
Authors’ response 1:
- Thank you for your valuable suggestion. We did our best to improve the completeness of the research by adding the background of the contents of this study to the introduction and verifying the research hypothesis. In response to your comment, we have revised the content as follows: ‘Aerosols generated during dental procedures have the potential to contaminate adjacent and distant sites [8]. Dental healthcare workers are exposed to the risk of viral infection through face-to-face communication, exposure to saliva, blood, and other body fluids, and handling of sharp dental instruments. They are exposed to aerosolised contaminants and airborne pathogens in the treatment room. In addition, since aerosols can stay suspended in the air for hours before settling on surrounding surfaces or entering the airway, viral infection can easily spread to larger areas via the aerosols exhaled from infected dental personnel and patients [9]’.
The explanation of the procedure carried out for the investigation needs to be improved, it is unclear, The section of statistical analysis, has to be worked more, much more
Authors’ response 2:
- We appreciate your concerns. We have revised the data collection schedule. In addition, the interpretation of statistical analysis results has been added, and all discrepancies between the results of tables and contents within the text have been cleared.
The discussion section must be improved and more worked, much more worked, there are hardly any lines, which are insufficient for a work to be published in our journal. In addition, you must connect with the introduction. This can be extended to the bibliography of the previous studies of the introduction that must be connected with the discussion, that will give more power to the manuscript,
Authors’ response 3:
- Thank you for this suggestion. We have revised and added more content to strengthen the discussion. Please refer to the section in the manuscript.
the same with the conclusions, are excessively brief
With these changes made, the article will improve and will have the quality to be published in this prestigious journal
Authors’ response 4:
- We appreciate the valuable suggestion. We have revised as follows: ‘In other words, dental hygienists’ burnout was found to increase as their SAVE and JS increased, and JS had a moderating effect in the relationship between SAVE and burn out.”
We have made our best efforts to accommodate your recommendations in the revised manuscript. Please let us know if you have any further recommendations for modifications. We would be glad to incorporate them. Thank you very much.

Reviewer 4 Report
Dear authors, thank you for the opportunity to get acquainted with your interesting work.
The article was written in compliance with all the rules of scientific presentation of the results of the study, an overview of modern research on the topic and an own empirical study on a representative sample were carried out.
At the same time, some sections in the work can be supplemented:
1. The section results of the study should contain not only tables, but also a description of the sequence of steps of statistical analyzes, for what purpose they were applied and briefly the results obtained. We see the description of the results so far only in the discussion of the results section.
2. The discussion section of the results needs to be expanded with a description of clearer directions for practical interventions from the point of view of the organization and from the point of view of the workers themselves. The statistically significant relationships obtained make it possible to do this.
The remarks made do not diminish the significance of your research and interest for readers.
Author Response
Thank you for reviewing our research. Below are our responses to your comments and queries. We have incorporated all your recommendations into the revised manuscript. Our revised paper has been checked by a native English speaker (American Journal Experts).
Dear authors, thank you for the opportunity to get acquainted with your interesting work.
The article was written in compliance with all the rules of scientific presentation of the results of the study, an overview of modern research on the topic and an own empirical study on a representative sample were carried out.
At the same time, some sections in the work can be supplemented:
- The section results of the study should contain not only tables, but also a description of the sequence of steps of statistical analyzes, for what purpose they were applied and briefly the results obtained. We see the description of the results so far only in the discussion of the results section.
Authors’ response 1:
- Thank you for your detailed assessment and suggestion. We added the results of a correlation analysis and hierarchical regression analysis and revised some interpretations.
- The discussion section of the results needs to be expanded with a description of clearer directions for practical interventions from the point of view of the organization and from the point of view of the workers themselves. The statistically significant relationships obtained make it possible to do this.
The remarks made do not diminish the significance of your research and interest for readers.
Authors’ response 2:
- We appreciate your suggestion. We've revised the content and added text to strengthen the discussion. Please refer to the Discussion section in the manuscript in red color.
Please let us know in detail if you have any further recommendations and we shall be glad to incorporate any further revisions required. Thank you very much.

Reviewer 5 Report
The results as presented are scarcely surprising or novel. I suspect that this paper constitutes a degree of 'salami slicing' in working a certain data set from another slightly different angle.
This is indicated by the frequent explanation of key psychometrics as being " ...as developed in previous studies..." or " ...as used in previous studies..."
This simply isn't good enough. The key psychometrics used in this study are not well known or even well validated NOR are they even described.
The statistical analysis may be fine but is largely meaningless if it is using key psychometrics of doubtful provenance.
So the results and their description and evaluation become automatically suspect.
In these circumstances I cannot support this paper
Author Response
Thank you for reviewing our research. Below are our responses to your comments and queries. We have incorporated all your recommendations into the revised manuscript. Our revised paper has been checked by a native English speaker (American Journal Experts).
The results as presented are scarcely surprising or novel. I suspect that this paper constitutes a degree of 'salami slicing' in working a certain data set from another slightly different angle.
Authors’ response 1:
- We appreciate your viewpoint. However, we would like to defend our research results by mentioning how this study explored the correlations among Stress and Anxiety to Viral Epidemics (SAVE), job stress (JS), and burnout among Korean dental hygienists during the COVID-19 pandemic and identified the moderating effect of JS. We believe that our study makes a significant contribution because there are few studies on the mental health of dental hygienists, especially during a pandemic situation. Moreover, our study revealed SAVE and JS to be significant influencing factors of burnout among dental hygienists. Further, we believe the findings of this study reveal the need for appropriate interventions to alleviate job-related stress and burnout among dental hygienists.
This is indicated by the frequent explanation of key psychometrics as being " ...as developed in previous studies..." or " ...as used in previous studies...“ This simply isn't good enough. The key psychometrics used in this study are not well known or even well validated NOR are they even described. The statistical analysis may be fine but is largely meaningless if it is using key psychometrics of doubtful provenance. So the results and their description and evaluation become automatically suspect. In these circumstances I cannot support this paper
Authors’ response 2:
- We appreciate your standpoint. The measurement variables in our study were used after confirming their validity based on previous studies. In response to your comment, and we did our best to revise the insufficient content in the manuscript in red color.
We have understood your concerns and tried to address them in the revised manuscript. Please let us know if any further modifications are needed, and we shall try to incorporate them. Thank you very much.

Round 2
Reviewer 3 Report
the authors have responded to my suggestions,
the article can be published
kind regards
Author Response
We sincerely appreciate your accept.
Reviewer 5 Report
Im afraid the authors have failed to address the important concerns I have about failing completely to identify and explain the key metrics used in the study beyond saying that they were 'derived from a previous paper'.
I should not have to read ' ...a previous paper...' to understand a current manuscript
Author Response
- We appreciate your opinion.
Despite the limitations you mentioned, it is a study of stress, anxiety and burn-out of dental hygienists in the current Covid-19 pandemic situation.
Therefore, We think it is a necessary study in the field of dental hygienics.
The discussion also described practical interventions and solutions in the dental and health sectors. In this respect, we can increase the reader's interest and importance in the study.
In addition, we revised and supplemented the introduction, method, results, and discussion so that many areas have been improved.
The points you commented on in this study will be reflected in after studies.
Thank you very much for your valuable opinion.
